# A Constrained Multi-Objective Reinforcement Learning Framework

**Sandy H. Huang**\*, **Abbas Abdolmaleki**\*, **Giulia Vezzani, Philemon Brakel,**
**Daniel J. Mankowitz, Michael Neunert, Steven Bohez, Yuval Tassa,**
**Nicolas Heess, Martin Riedmiller, Raia Hadsell**
DeepMind
London, UK
`shhuang, aabdolmaleki@deepmind.com`

**Abstract:** Many real-world problems, especially in robotics, require that reinforcement learning (RL) agents learn policies that not only maximize an environment reward, but also satisfy constraints. We propose a high-level framework for solving such problems, that treats the environment reward and costs as separate objectives, and learns what preference over objectives the policy should optimize for in order to meet the constraints. We call this Learning Preferences and Policies in Parallel (LP3). By making different choices for how to learn the preference and how to optimize for the policy given the preference, we can obtain existing approaches (e.g., Lagrangian relaxation) and derive novel approaches that lead to better performance. One of these is an algorithm that learns a *set* of constraint-satisfying policies, useful for when we do not know the exact constraint a priori.

**Keywords:** constrained RL, multi-objective RL, deep RL

## 1 Introduction

Many problems, especially in the real world and in robot applications, require that policies meet constraints [1]. For instance, we might want a factory robot to optimize task throughput while keeping actuator forces below a threshold, to limit wear-and-tear. Or, we might want to minimize energy usage for cooling a data center while ensuring temperatures remain below some level [2].

Existing approaches typically formulate such constrained RL problems as a constrained Markov Decision Process (CMDP) [3]. To solve it, many rely on Lagrangian relaxation [4], which transforms the constrained optimization to be unconstrained, where the objective is a sum of the expected task return and weighted constraint functions. The weights are updated to ensure constraint satisfaction.

However, some constrained RL problems do not fit the CMDP formulation. For instance, we might not have an exact constraint threshold in mind, but rather a viable range. Or, we may have *soft constraints* [5], where violating the constraint is undesirable but not catastrophic. These situations arise because of the inherent trade-off between maximizing task reward and satisfying constraints.

Our key insight is to view constrained RL from a multi-objective perspective, where the task reward and each constraint are separate objectives. To train a policy, we must tell it what preference over objectives to optimize for—i.e., how much to prioritize one objective over another. At the extremes, we could tell the policy to either focus entirely on maximizing task reward while disregarding constraint satisfaction, or focus entirely on satisfying the constraints. Some preference settings will lead to policies that satisfy the constraints, but we do not know in advance which ones these are.

Thus we propose to learn which preferences lead to constraint-satisfying policies, while simultaneously training policies for those preferences. We call this framework ***Learning Preferences and Policies in Parallel*** (**LP3**), shown in Fig. 1. In LP3 there are two main design choices: 1) which multi-objective RL algorithm is used to train the policy, and 2) whether to learn a single preference or a distribution over preferences. Lagrangian relaxation is one possible instantiation—it uses linear scalarization as the multi-objective RL algorithm and learns a single preference (Sec. 4.2).

---

\*equal contribution

5th Conference on Robot Learning (CoRL 2021), London, UK.

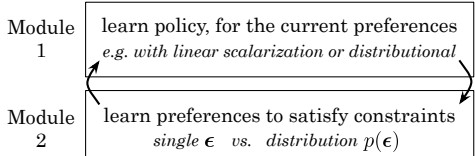

| Algorithm | Module 1 | Module 2 |
|---|---|---|
| Lagrangian relaxation | linear scalarization | single |
| LP3 [MO-MPO] | distributional (MO-MPO) | single |
| LP3 [MO-MPO-D] | distributional (MO-MPO) | distribution |

Figure 1: We use our proposed framework (left), Learning Preferences and Policies in Parallel (LP3), to derive two novel algorithms for constrained RL by making different design choices (in italics).

We explore two other instantiations of our framework. In the first, to train the policy we use a state-of-the-art multi-objective RL algorithm, Multi-Objective Maximum a Posteriori Optimization (MO-MPO) [6]. In the second, we learn a probability distribution over preferences, based on how likely they are to lead to constraint-satisfying policies. This enables finding a *set* of constraint-satisfying policies in a single training run, that differ in terms of task return and constraint performance.

Our main contributions, among others, are:

- We propose a multi-objective framework for constrained RL, called LP3 (Sec. 4). This is a general framework, that includes Lagrangian relaxation as one possible instantiation. Although the connection between constrained and multi-objective problems is intuitive [7, 8], our work is the first to leverage this to create novel algorithms for constrained RL.
- Based on this framework, we derive two novel algorithms for constrained RL. One of these is the first algorithm, to the best of our knowledge, for finding a set of constraint-satisfying policies—this is important when the constraint is soft or not exactly known a priori.
- We run extensive experiments that compare these two new approaches with Lagrangian-based approaches on continuous control tasks across three task suites, including high-dimensional locomotion tasks. Videos are at http://sites.google.com/view/cmorl.

## 2  Related Work

**Constrained reinforcement learning.**   Constrained RL algorithms learn policies that satisfy constraints while maximizing task return. Many approaches formulate this problem as a CMDP and use Lagrangian relaxation [4]. Recent works claim this converges to constraint-satisfying [9] or optimal [10] solutions. Other works seek to stabilize this optimization by incorporating convex relaxations [11] or derivatives of the constraint function [12]. Others have applied Lagrangian relaxation to mean-value constraints [9], convex set constraints [13], and local constraint satisfaction [14].

For soft constraints, MetaL [5] uses meta-gradients to adjust the Lagrangian learning rate, to find a single policy that achieves high task reward without overly violating the constraints. However, this particular trade-off between the reward and cost may not be desirable. Instead, we propose an approach that returns a *set* of policies to choose from, that make different trade-offs.

**Safe reinforcement learning.**   In safe RL, the aim is to achieve constraint satisfaction not only at deployment, but also during learning [15]. Recent works for deep RL modify the policy improvement step to guarantee that the policy will never violate constraints during training [16, 17, 18, 19, 20, 21]. These approaches require, however, that the initial policy (nearly) satisfies the constraints; otherwise, performance may decrease significantly [22]. We focus on constrained RL, rather than safe RL, but we discuss how to extend our approach to reduce constraint violation during training (Sec. 6).

**Multi-objective reinforcement learning.**    In our framework, the policy is trained with multi-objective RL (MORL). MORL consists of single-policy and multi-policy approaches. Single policy methods learn a policy that is optimal for a given trade-off. Most rely on linear scalarization [23], which restricts solutions to the convex portions of the Pareto front and can be sensitive to reward scales. Non-linear scalarizations exist [24, 25, 26], but these are harder to combine with value-based RL and have seen limited use in deep RL. Recently, Abdolmaleki et al. [6] introduced MO-MPO, a single-policy approach that does not rely on scalarization and is invariant to reward scales.

Multi-policy MORL aims to find a *set* of policies that covers the whole Pareto front. Recent works learn a manifold in parameter space [27, 28] or combine single policy approaches with a general objective [29], to directly optimize the hypervolume of the Pareto front.

# 3 Background and Notation

## 3.1 Markov Decision Processes

A Markov Decision Process (MDP) is defined by $\{\mathcal{S}, \mathcal{A}, p, r_0, \gamma\}$: the state space $\mathcal{S}$, action space $\mathcal{A}$, transition function $p(s'|s, a)$, bounded task reward function $r_0(s, a)$, and discount factor $\gamma \in [0, 1]$.

**Constrained MDP.** Constrained RL problems are typically formulated as constrained MDPs (CMDP). A CMDP is defined by $\{\mathcal{S}, \mathcal{A}, p, r_{0:K}, \beta_{1:K}, \gamma\}$. Compared to a regular MDP, a CMDP includes additional constraint-specific reward functions $\{r_k(s, a)\}_{k=1}^K$ with corresponding thresholds $\{\beta_k\}_{k=1}^K$. All reward functions $r_k$ return scalar rewards; we will refer to these as separate *objectives*.

A policy $\pi(a|s)$ maps from a state to a distribution over actions. Let $J_k^\pi = \mathbb{E}_\pi[\sum_t \gamma^t r_k(s_t, a_t)]$ denote the expected return for objective $k$ when following policy $\pi$, given a fixed initial state distribution. The action-value function $Q_k^\pi(s, a)$ is the expected return for objective $k$ after taking action $a$ in state $s$, and then acting based on policy $\pi$.

The optimal solution to a CMDP is a policy that maximizes expected task return, while ensuring all other expected returns satisfy their thresholds:

$$\max_\pi \ J_0^\pi \quad \text{s.t. } J_k^\pi \geq \beta_k \ \ \forall k = 1, \ldots, K . \tag{1}$$

Many approaches for solving CMDPs are based on Lagrangian relaxation, which turns the constrained problem into the following unconstrained optimization problem:

$$\min_{\lambda \geq 0} \max_\pi \ J_0^\pi + \sum_{k=1}^K \lambda_k \left( J_k^\pi - \beta_k \right) . \tag{2}$$

These approaches alternate between optimizing the Lagrange multipliers $\lambda_k$ and the policy $\pi$.

**Multi-Objective MDP.** A multi-objective MDP (MO-MDP) is defined by $\{\mathcal{S}, \mathcal{A}, p, r_{0:K}, \gamma\}$. In contrast to an (C)MDP, there are multiple rewards to optimize for, and no constraints. The optimal policy depends on the desired preference $\epsilon$ across objectives. In *linear scalarization*, the preference $\epsilon$ defines a weighting across objectives; the optimal policy is the one that maximizes $\sum_{k=0}^K \epsilon_k J_k^\pi$.

When no preferences are given, the solution to a MO-MDP is the set of all non-dominated policies, also known as the *Pareto front*. A policy is *non-dominated* if there is no other policy that improves its return for one objective without decreasing return for another.

## 3.2 Multi-Objective MPO (MO-MPO)

Whereas most multi-objective RL algorithms use linear scalarization, MO-MPO [6] takes a distributional approach to combining objectives. MO-MPO extends MPO [30] to the multi-objective setting. MO-MPO is a policy iteration algorithm with two steps: policy evaluation and policy improvement.

**Policy evaluation.** In this step, MO-MPO evaluates the current policy $\pi_{\text{old}}(a|s)$ by training a Q-function per objective, as according to Q-decomposition [31].

**Policy improvement.** Given the previous policy $\pi_{\text{old}}(a|s)$ and Q-functions $\{Q_k^{\text{old}}(s, a)\}_{k=0}^K$, in this step MO-MPO improves the policy for a given state visitation distribution $\mu(s)$. MO-MPO decomposes policy improvement into two steps, similar to the view proposed in Ghosh et al. [32]: first find per-objective improved action distributions, and then distill these into a new policy.

*Finding per-objective distributions:* For each objective $k$, MO-MPO computes a nonparametric policy $q_k(a|s)$ that improves the old policy with respect to that objective, subject to a non-negative constraint $\epsilon_k$ on the KL-divergence between the improved and old policies:

$$\max_{q_k} \ \mathbb{E}_{\mu(s)}\left[ \int_a q_k(a|s) \, Q_k^{\text{old}}(s, a) \, \mathrm{d}a \right] \quad \text{s.t. } \mathbb{E}_{\mu(s)}\left[ \text{KL}(q_k(\cdot|s) \,\|\, \pi_{\text{old}}(\cdot|s)) \right] < \epsilon_k . \tag{3}$$

*Learning a new parameterized policy:* MO-MPO then updates the policy via supervised learning on these nonparametric policies, subject to a trust region with bound $\beta > 0$ for more stable learning:

$$\min_\theta \ \sum_{k=0}^N \mathbb{E}_{\mu(s)}\left[ \text{KL}(q_k(\cdot|s) \,\|\, \pi_\theta(\cdot|s)) \right] \quad \text{s.t. } \mathbb{E}_{\mu(s)}\left[ \text{KL}(\pi_{\text{old}}(\cdot|s) \,\|\, \pi_\theta(\cdot|s)) \right] < \beta . \tag{4}$$

The KL-constraints $\{\epsilon_k\}_{k=0}^K$ encode preferences: the larger $\epsilon_k$, the more influence objective $k$ has.

# 4 Learning Preferences and Policies in Parallel (LP3)

In this section we introduce LP3, our constrained multi-objective RL framework. We first formulate the constrained RL problem as a Constrained Multi-Objective MDP (CMO-MDP). Following this, we describe how LP3 solves this MDP, and detail how Lagrangian relaxation algorithms are an instantiation of this framework. We then derive two novel constrained RL algorithms based on LP3.

## 4.1 Constrained Multi-Objective MDP (CMO-MDP)

The CMDP formulation implies a single optimal solution. But in real-world settings, one may be considering a *range* of viable constraint thresholds, which correspond to different trade-offs between the task reward and constraints. This connects to the multi-objective setting.

We define a constrained multi-objective MDP (CMO-MDP) as $\{\mathcal{S}, \mathcal{A}, p, r_{0:K}, \beta_{N:K}, \gamma\}$, where $N \geq 1$. In a CMO-MDP, the optimization is over $K$ reward functions, where $N$ of these are task rewards, and the remaining $K - N$ rewards have constraints on them. Thus a CMO-MDP differs from a CMDP in two key ways: 1) the optimization objective is over *all* rewards, including the constraint-specific ones; and 2) there may be multiple task rewards $r_{0:N}$.

A CMO-MDP compares constraint-satisfying policies in terms of their returns for the task reward *and* constraint functions—its solution is the set of all constraint-satisfying Pareto optimal policies. In contrast, a CMDP ranks policies according to only task return—its solution is the single constraint-satisfying policy with the highest task return.

## 4.2 The LP3 Framework for Solving CMO-MDPs

We now describe LP3, a unifying framework (Fig. 1) for deriving algorithms that solve CMDPs and CMO-MDPs. This framework has two modules: learning the policy, and learning which preference(s) (over the task and constraints) the policy should optimize for to meet the constraints.

For the first module, the key choice is which multi-objective RL algorithm (such as linear scalarization or MO-MPO) is used to train the policy $\pi_\theta$ to optimize for a given preference.

For the second module, the key choice is whether to learn a single preference $\epsilon$ or a preference distribution $p(\epsilon)$. The former corresponds to learning a policy $\pi_\theta(a|s)$. Learning a distribution over preferences, however, requires training a preference-conditioned policy $\pi_\theta(a|s, \epsilon)$, that learns to optimize for different preference settings. This enables finding a set of constraint-satisfying policies, produced by sampling preferences from the final $p(\epsilon)$ and rolling out the fully-trained $\pi_\theta(a|s, \epsilon)$, conditioned on that preference. In practice, using a preference distribution can also improve learning, because training over multiple preferences enables more diverse data collection and exploration.

Note that we can use different time scales for training the two modules in this framework. In our experiments, we alternate between taking one learning step on each.

**Connection to Lagrangian Relaxation.** As mentioned, Lagrangian relaxation methods are equivalent to LP3 with linear scalarization and a single preference optimized using gradient descent. To show this, define the preference $\epsilon = [1, \lambda_1, \lambda_2, \ldots, \lambda_K]^\top$, where $\lambda_{1:K}$ are the Lagrange multipliers.

In the first module, we train the policy to maximize a linear combination of Q-values, weighted by the current preferences $\epsilon^{\text{old}}$: $\sum_{k=0}^{K} \epsilon_k^{\text{old}} Q_k(a, s)$. We can use any standard RL algorithm for this.

In the second module, given the current policy $\pi_{\text{old}}(a|s)$, we use gradient descent to optimize for a single preference setting $\epsilon$ with respect to the loss $\mathcal{L}(\epsilon > 0) = \sum_{k=1}^{K} \epsilon_k f_k$. Here, $f_k$ is the fitness score corresponding to whether the policy respects threshold $\beta_k$:

$$f_k = \mathbb{E}_{\mu(s)} \left[ Q_k^{\text{old}}(s, \mathbb{E}_{\pi_{\text{old}}(a|s)}[a]) \right] - \beta_k \, . \tag{5}$$

## 4.3 Novel Instantiations of LP3

Making different design choices in the LP3 framework leads to two novel algorithms for constrained RL. The first replaces linear scalarization with a state-of-the-art multi-objective RL algorithm, MO-MPO. The motivation is that MO-MPO may be better at training policies for a given preference,

since it has been shown to outperform linear scalarization on a variety of control problems [6]. We call this ***LP3 [MO-MPO]***. It uses the same second module (for learning the preference) as Lagrangian relaxation. The second algorithm, in addition to using MO-MPO, learns a distribution over preferences. We call this ***LP3 [MO-MPO-D]*** and describe it below; full details are in Appendix E.

### 4.3.1 Module 1: Preference-Conditioned MO-MPO

Learning a distribution over preferences requires training a preference-conditioned policy $\pi_\theta(a|s, \epsilon)$, but MO-MPO can only train a policy $\pi_\theta(a|s)$ for a single preference $\epsilon$. We extend MO-MPO to train a *single* policy $\pi_\theta(a|s, \epsilon)$ conditioned on preferences $\epsilon \sim p_{\text{old}}(\epsilon)$, by making both the policy evaluation and policy improvement steps preference-conditioned as follows.

**Preference-conditioned policy evaluation.** We evaluate the current policy $p_{\text{old}}(\epsilon)\pi_{\text{old}}(a|s, \epsilon)$ via off-policy learning of per-objective Q-functions $Q_k^\pi(a, s, \epsilon)$. Note that these Q-functions must be preference-conditioned because the policy is. For each learning step we sample $L$ transitions from the replay buffer, $\{s_i, a_i, \{r_i^k\}_k^K, s_i'\}_i^L$. Motivated by prior work on learning from off-policy data [33], we use relabeling to augment the states with preferences sampled from the current preference distribution $\epsilon_i \sim p_{\text{old}}(\epsilon)$, resulting in transitions $\{[s_i, \epsilon_i], a_i, \{r_i^k\}_k^K, [s_i', \epsilon_i]\}_i^L$. Any policy evaluation algorithm can be used. We use distributional policy evaluation with 5-step return [34].

**Preference-conditioned policy improvement.** This step improves the current policy $\pi_{\text{old}}(a|s, \epsilon)$.

*Finding per-objective distributions:* To find preference-conditioned per-objective improved action distributions $q_k(a|s, \epsilon)$, we optimize the following policy optimization problem for each objective:

$$\max_{q_k} \quad \mathbb{E}_{p_{\text{old}}(\epsilon)\mu(s)}\Big[ \int_a q_k(a|s, \epsilon)\, Q_k^{\text{old}}(s, a, \epsilon)\, \mathrm{d}a \Big] \tag{6}$$

$$\text{s.t.} \quad \mathbb{E}_{\mu(s)}\Big[\text{KL}(q_k(\cdot|s, \epsilon)\|\pi_{\text{old}}(\cdot|s, \epsilon))\Big] < \epsilon_k \quad \forall\, \epsilon \sim p_{\text{old}}(\epsilon)\,.$$

This generalizes (3) from MO-MPO, which assumes fixed KL-constraints $\epsilon$, to a distribution $p_{\text{old}}(\epsilon)$ over KL-constraints. We use a closed-form known nonparametric solution for $q_k$ [35, 30].

*Learning a new parameterized policy:* After obtaining per-objective improved policies, we use supervised learning to distill these into a new parameterized preference-conditioned policy:

$$\min_\theta \sum_{k=0}^K \mathbb{E}_{p_{\text{old}}(\epsilon)\mu(s)}\big[\text{KL}(q_k(\cdot|s, \epsilon)\|\pi_\theta(\cdot|s, \epsilon))\big] \quad \text{s.t.}\ \mathbb{E}_{p_{\text{old}}(\epsilon)\mu(s)}\big[\text{KL}(\pi_{\text{old}}\|\pi_\theta)\big] < \beta. \tag{7}$$

### 4.3.2 Module 2: Learning Preferences

The second module optimizes the preference $\epsilon$ or preference distribution $p(\epsilon)$ to lead to better constraint satisfaction. This assumes fitness functions $f_k$ that evaluate satisfaction of the constraint threshold $c_k$, given the current policy $\pi_{\text{old}}$ and Q-function $Q_k^{\text{old}}$.[2] We can use any gradient-based algorithm to optimize the preferences, initialized by $\epsilon_{\text{old}}$ or $p_{\text{old}}(\epsilon)$, to maximize the fitness.

In our empirical evaluation, we use fitness functions that give a score of zero to preferences that lead to constraint satisfaction, and negative scores to all others. For equality constraints we use a fitness function that penalizes the difference between the expected Q-values and constraint threshold, and for inequality constraints it penalizes the amount the expected Q-values violate the threshold:

$$f_k^{\text{eq}}(\epsilon) = -\Big|\mathbb{E}_{\mu(s)}\Big[Q_k^{\text{old}}(s, \mathbb{E}_{\pi_{\text{old}}}[a], \epsilon)\Big] - \beta_k\Big| \tag{8}$$

$$f_k^{\text{in}}(\epsilon) = \min\Big(0,\ \mathbb{E}_{\mu(s)}\Big[Q_k^{\text{old}}(s, \mathbb{E}_{\pi_{\text{old}}}[a], \epsilon)\Big] - \beta_k\Big). \tag{9}$$

## 5 Experiments

To recap, in the LP3 framework, Lagrangian relaxation corresponds to using linear scalarization (LS) and learning a single preference (Fig. 1). Lagrangian relaxation is a widely-used approach for

---

[2]The flexibility in choosing the fitness function enables specifying a variety of constraints: e.g., expected return equal to a threshold, within an interval of thresholds, or within the union of disjoint threshold intervals.

Figure 2: In a toy domain, we trained policies for four different constraints on $r_2$ per task. When the Pareto front is concave (right), the LP3 [LS] baseline cannot recover the optimal solutions, whereas LP3 [MO-MPO] can. This is because LP3 [LS] relies on linear scalarization (LS), which fundamentally cannot find solutions on concave Pareto fronts [36].

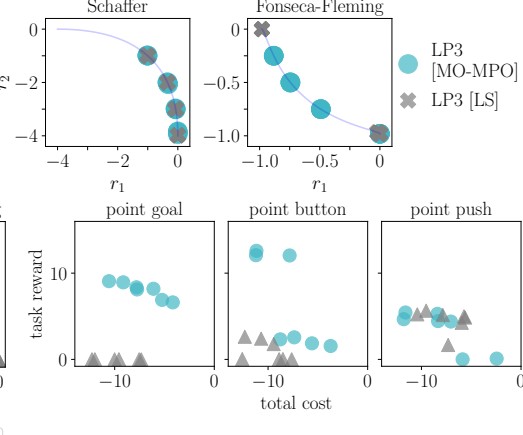

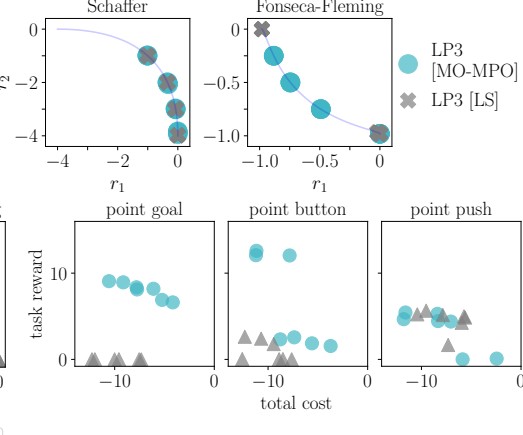

Figure 3: Using MO-MPO (circles) for Module 1 results in better policies than using LS (triangles), for hard-to-satisfy constraints. These plots show the performance of policies trained on different constraint thresholds. Each point denotes the task reward and cost of a fully-trained policy; points above and to the right are better. The LP3 [LS] baseline struggles to find policies that perform well on both the task and cost. LP3 [LS] also cannot find solutions on a concave Pareto front (humanoid walk) and takes longer to learn the task (run, mid-training).

constrained RL because of its simplicity and effectiveness [22, 12]. We proposed two novel changes: 1) using MO-MPO instead of LS in Module 1, and 2) learning a distribution over preferences rather than a single preference in Module 2. In our empirical evaluation, we first run ablation studies that study the impact of each of these changes. Then, we run further experiments with LP3 [MO-MPO-D], that combines both improvements. The main baseline we compare against is **LP3 [LS]**, which uses MPO [30] to optimize the Lagrangian dual objective (2). This is a strong Lagrangian-based baseline that obtains state-of-the-art results on benchmark constrained RL tasks (Appendix C.2).

**Implementation.** We train Gaussian policies parameterized by neural networks. Our implementations are based on the open-sourced MPO and MO-MPO in Acme [37]. For LP3 [MO-MPO-D], we learn a separate distribution per preference $\epsilon_k$, initialized to uniform. We choose to use a discrete distribution because it is flexible and interpretable. Unless otherwise mentioned, we use (9) as the fitness function for LP3 [MO-MPO] and use (8) for LP3 [MO-MPO-D]—so that it converges to a single policy, rather than a set of policies, for easier comparison with the single-preference approaches. We sweep over hyperparameters for each approach; details are in Appendix A.

### 5.1 Module 1: Linear Scalarization vs. MO-MPO

To first assess the impact of using MO-MPO instead of linear scalarization, we compare LP3 [MO-MPO] against LP3 [LS] across several domains. LP3 [LS] is an ablation of LP3 [MO-MPO] with respect to Module 1; both learn a single preference in the same way for Module 2.

**Toy Domain.** First, we consider a toy domain with a continuous action ($a \in \mathbf{R}$) and two objectives ($r(a) \in \mathbf{R}^2$). The constraint is on $r_2(a)$. For the reward, we use either the Schaffer or Fonseca-Fleming function, which are standard test functions in multi-objective optimization [38]. For each function, we train policies for four constraint thresholds. For Schaffer, both approaches find the optimal solutions. However, for Fonseca-Fleming, which has a concave Pareto front, only LP3 [MO-MPO] can find all optimal solutions (Fig. 2, right). LP3 [LS] only finds solutions at the two extremes, even after tuning the learning rate. This is likely because linear scalarization cannot find solutions on the concave portions of a Pareto front [36]. Even in this simple setting, we can see the potential advantages of using MO-MPO over linear scalarization, in Module 1 of LP3.

**Humanoid.** We next consider two high-dimensional locomotion tasks, humanoid run and walk from the DeepMind Control Suite [39]. The constraint is imposed on the expected negative control norm (i.e., $-\|a\|_2$), which roughly captures the "energy" expended by the agent.

We train policies using LP3 [LS] and LP3 [MO-MPO] for a variety of constraint thresholds. We plot the average task reward and cost for the fully-trained policies in Fig. 3. LP3 [MO-MPO] finds higher-quality solutions for both tasks, especially for walk, which may be because its ground-truth

| Task | Algorithm | Task | Cost | Task | Algorithm | Task | Cost |
|---|---|---|---|---|---|---|---|
| cartpole swingup | MetaL | $850 \pm 1$* | $-342 \pm 1$* | quadruped walk | MetaL | $999 \pm 0$* | $-615 \pm 15$* |
| | D4PG-Lag | $376 \pm 41$* | $-199 \pm 29$* | | D4PG-Lag | $999 \pm 0$* | $-632 \pm 14$* |
| | LP3 [LS] | $166 \pm 20$* | $-293 \pm 108^{\dagger}$ | | LP3 [LS] | $\mathbf{1000 \pm 0}$ | $-33 \pm 8$* |
| | LP3 [MO-MPO-D] | $\mathbf{871 \pm 4}$ | $\mathbf{-80 \pm 4}$ | | LP3 [MO-MPO-D] | $\mathbf{1000 \pm 0}$ | $\mathbf{0 \pm 0}$ |
| humanoid walk | MetaL | $659 \pm 54$* | $-368.3 \pm 26$* | walker walk | MetaL | $966 \pm 0$* | $-164.1 \pm 9$* |
| | D4PG-Lag | $571 \pm 53$* | $-361.3 \pm 19$* | | D4PG-Lag | $905 \pm 11$* | $\mathbf{-76.8 \pm 3}$ |
| | LP3 [LS] | $\mathbf{668 \pm 149}$ | $-104.3 \pm 41^{\dagger}$ | | LP3 [LS] | $\mathbf{983 \pm 1}$ | $-58 \pm 5$ |
| | LP3 [MO-MPO-D] | $828 \pm 28$ | $\mathbf{-32 \pm 3}$ | | LP3 [MO-MPO-D] | $\mathbf{983 \pm 0}$ | $-61 \pm 28$ |

Table 1: On RWRL tasks, LP3 [MO-MPO-D] achieves better task reward *and* cost than other algorithms. The bounds are on standard error. Results for MetaL and D4PG-Lag are from [5]; both rely on linear scalarization. * ($p \leq 0.05$) and † ($p \leq 0.1$) indicate significance based on Welch's t-test, compared to LP3 [MO-MPO-D].

Pareto front is concave. LP3 [MO-MPO] is also more sample efficient: for humanoid run, mid-way through training LP3 [MO-MPO] policies obtain moderate task reward, while LP3 [LS] policies have not learned the task at all (Fig. 3, "run, mid-training"). This could be because LP3 [MO-MPO] can optimize for both objectives at once, whereas LP3 [LS] cannot (see analysis in Appendix C.4).

### 5.2   Module 2: Single Preference vs. Distribution over Preferences

We combine MO-MPO, because it outperforms linear scalarization, with learning a distribution over preferences to obtain LP3 [MO-MPO-D]. LP3 [MO-MPO] is an ablation of LP3 [MO-MPO-D] with respect to Module 2. For humanoid, both approaches perform similarly (Fig. 3).

A key advantage of using LP3 [MO-MPO-D], as opposed to single-preference approaches, is the ability to solve a CMO-MDP and discover a *set* of constraint-satisfying non-dominated solutions, rather than finding a single solution. This is useful because in practice, we may not know the true constraint ahead of time, and would thus prefer to see several options along the Pareto front. We evaluate this in the following section on a task with three objectives, two of which are constrained.

### 5.3   Evaluation of Full Algorithm

Finally, we focus on comparing LP3 [MO-MPO-D], which combines both improvements, against existing approaches in the literature, as well as our LP3 [LS], a strong Lagrangian-based approach.

**RWRL.**   We use the same four Real World RL (RWRL) [40] tasks and constraints used by Calian et al. [5], and we compare against the algorithms evaluated in that work (described in Appendix B.5). The cost is binary and depends on the task, e.g., for humanoid walk, the cost is $-1$ if the joint angles are outside of a specified range, and 0 otherwise. For each task, we used LP3 [MO-MPO-D] and LP3 [LS] to train five policies with different random initializations. LP3[MO-MPO-D] performs best on three out of four tasks, incurring less cost while achieving on-par or better task reward (Table 1).

**Safety Gym.**   From OpenAI Safety Gym [22], we use the Level 2 point mass tasks: goal, button, and push. There is a sparse reward for either reaching a goal location or pushing a box to a goal. There is a cost for running into or over objects. On two out of the three tasks, LP3 [MO-MPO-D] finds better solutions than LP3 [LS], on harder-to-satisfy constraint thresholds (Fig. 3). For point push and for easier-to-satisfy constraint thresholds, LP3 [LS] performs on par with LP3 [MO-MPO-D] (see Appendix C). We hypothesize this is because when meeting the constraint does not conflict much with the task objective, it is easier to find high-task-reward solutions that incur small cost.

**Robot.**   We created a simulated robot manipulation task with the Sawyer arm, with a sparse reward for reaching a green block and pushing it to a target position. There is a cost for either the gripper or green block colliding with the yellow block (Fig. 4, left). On this task, LP3 [MO-MPO-D] and LP3 [LS] converge to similar task performance and constraint satisfaction, but LP3 [LS] takes about twice as many actor steps to do so (Fig. 4). This is likely due to the use of MO-MPO in Module 1—recall that in the first ablation study, this improved sample efficiency in humanoid run (Sec. 5.1).

**Multiple Constraints.**   We also evaluate on tasks with three and four objectives: walker run and humanoid walk from the Control Suite. In walker run, there are three objectives: a torso height reward, a moving forward reward, and a cost for energy usage (captured by negative control norm). In humanoid walk, there are four objectives: the task reward (from the standard Control Suite task), a moving forward reward, a moving left reward, and a cost for energy usage.

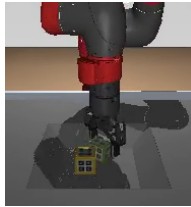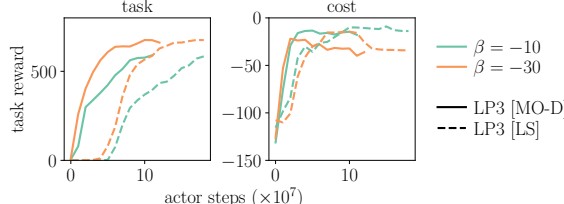

Figure 4: On the robot push task, LP3 [LS] is much less sample-efficient: it takes about twice as many actor steps to converge to similar task and cost performance as LP3 [MO-MPO-D]. $\beta$ denotes the constraint threshold.

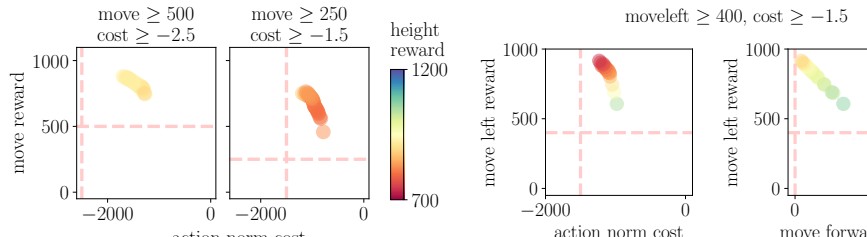

Figure 5: For walker run (left) and humanoid walk (right), we use LP3 [MO-MPO-D] to train policies that learn a portion of the Pareto front. Walker has three objectives and humanoid has four objectives; both tasks have inequality constraints on two objectives. These plots shows the Pareto front for one random seed; Appendix C.6 contains plots for two others. The dashed lines indicate the constraint threshold. We evaluate each fully-trained policy by conditioning on learned preferences: for each objective with a constraint, we select preferences at evenly-spaced percentiles from its learned distribution and use the median preference for the other objectives. Since we can only plot two objectives at a time, we use color to indicate the value of another. For humanoid walk, the not-plotted fourth objective (task reward) has near-maximum reward regardless of the preference.

For both tasks, we set inequality constraints on moving and energy usage. We use LP3 [MO-MPO-D] to train preference-conditioned policies using the fitness function (9). These policies learn a portion of the Pareto front that satisfies the constraints (Fig. 5), which is useful when the constraint is soft or not known exactly a piori. For humanoid walk, the agent learns to smoothly interpolate between walking left and walking forward (Fig. 5, right-most plot). In this task, walking left is a more challenging gait and would not likely emerge naturally without a constraint on this objective.

Appendix C contains further experiments and analysis, including on constraint satisfaction (Appendix C.5) and evaluation with multiple random seeds per constraint threshold (Appendix C.7).

## 6   Conclusion and Future Work

In this paper, we proposed to formulate constrained RL as a constrained multi-objective MDP. Based on this multi-objective perspective, we described a general framework, LP3, for solving constrained RL problems. We used this perspective to introduce two new algorithms: 1) using a better multi-objective RL algorithm, MO-MPO, instead of linear scalarization; and 2) learning a distribution over preferences instead of a single preference, to find a set of constraint-satisfying non-dominated policies. For safe RL, one could initialize the preference distribution to prioritize minimizing cost.

This work has several limitations. One is that LP3 [MO-MPO(-D)] has not been validated yet on a real robot platform. However, MPO with Lagrangian relaxation has been successfully run on robots [e.g. 14], and our ablation study suggests LP3 [MO-MPO] should perform at least as well. Another limitation is that LP3 depends on accurate policy evaluation to ensure constraint satisfaction, because it relies on Q-value estimates to learn preferences. The preference learning rate cannot be too high, otherwise the distribution will converge based on Q-values estimates for a partially-trained policy—which may not accurately reflect which preferences will lead to constraint satisfaction for the fully-trained policy. In addition, using a discrete distribution for the preferences requires that its support includes preferences that satisfy the constraint threshold. Finally, a limitation is that we focus on problems with up to four objectives. In principle, LP3 [MO-MPO-D] can be scaled up to more objectives with minimal additional computational cost, and this can be easily parallelized. However, the more objectives there are that are in conflict with each other, the more difficult the problem is. It would be valuable to investigate performance on these problems in future work.

**Acknowledgments**

The authors would like to thank Dan Calian, Andrea Huber, Leslie Fritz, Christopher Schuster, the Control Team, the REAL Team, and many others at DeepMind for their helpful feedback and support for this paper. We would also like to thank our anonymous reviewers for their valuable feedback.

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
