# OpenReview forum: "A Constrained Multi-Objective Reinforcement Learning Framework"
_robot-learning.org/CoRL/2021/Conference — CoRL2021 Poster_

### Official Review · Reviewer_rQa3 · 2021-07-20

**Originality:** Good
**Technical Quality:** Very Good
**Clarity Of Presentation:** Excellent
**Impact:** 3

**Recommendation:**

Weak Accept: I recommend accepting the paper, but will not argue for my recommendation if the majority of other reviewers have a different opinion.

**Summary:**

This work proposes a general framework to leverage multi-objective learning for constraint RL, for which Lagrangian relaxation is one possible option. Further, the authors propose two advances to this setting. First, instead of linear scalarization they use Multi-Objective Maximum-a-Posteriori Optimization (MO-MPO) to allow for a distributional approach that combines objectives. Second, they introduce the use of a preference distribution in place of a single preference. This additionally enables to find a set of constraint-satisfying policies that offer different levels of trade-off between constraints and task performance.

**Issues:**

### Minor Issues
- One of the main references is cited incorrectly: D. A. Calian, D. J. Mankowitz, T. Zahavy, Z. Xu, J. Oh, N. Levine, and T. Mann. Balancing
constraints and rewards with meta-gradient d4pg. arXiv preprint arXiv:2010.06324, 2020. --> ICLR 2021
- Table 1: LP3 [LS] Humanoid walk is neither bold nor marked with a $p\leq0.05$ for Welch's t-test
- Comparing the numbers reported in this work and the numbers reported in table 7 by Calian, et al. I cannot find a direct match, most of the results are close but not identical.

**Reviewer Expertise:**

Fair: Some knowledge of the area

**Strengths And Weaknesses:**

Using multi-objective optimization for constraint RL is an intuitive direction. While the idea in itself is not novel, this work is still introducing the idea to a new area and is at the same time approaching an important field in deep RL and Robotics.
Overall, this work is well motivated, structured excellently, and provides sufficient background information to understand all aspects presented throughout every chapter. I very much enjoyed reading this paper and the general approach itself was apparent.

An important weakness with the current approach is the applicability to physical robotics systems, which the authors state in their conclusion as interest for future work. However, given the currently required sample complexity, it might be challenging to directly apply the algorithm in its current form. Other than that, the work does not put much emphasis on demonstrating the limitations of the framework. Only the appendix contains additional discussion, e.g. for the `point` task the constraints are consistently violated due to the Q-function approximation.

**Summary Of Recommendation:**

First and foremost, the technical quality as well as the clarity of this work is well done and there is little to complain about.
The novelty of the proposed framework is mainly given by transferring the existing idea of multi-objective learning for constraint optimization to the (deep) RL space. While this is not a groundbreaking idea, I think it has its place in the given setting.

Despite the good writing, I am still left with some open points:
- For the preference-conditioned MO-MPO, the replay buffer transitions are relabeled based on the current preference distribution $p(\epsilon)$. Do you have any results that demonstrate that this indeed leads to improved stability and sample efficiency?
- While the approach can learn a set of constraint-satisfying policies, how would one choose an appropriate trade-off. In simulation one option would compare multiple policies, albeit, this is assuming a human expert that can search the preference distribution. Considering a physical system, this might not even be an option, especially when trying to avoid potentially system-harming/-damaging trade-offs.
- For easier comparison LP3 [MO-MPO-D] uses equality constraints to find a single solution. However, wouldn't that lead to a different optimization problem in the first place?
- Figure 4 shows how the learned set performs, when using inequality constraints, albeit to me it seems the coverage of the Pareto front is quite seed dependent. Exploring this more would also be in the interest of determining the limitations of the framework.
- For the toy domain I did not find any details about the initialization or seed choice. It might as well be that LP3 [LS] is performing poor simply because of the chosen seed and not necessarily due to the concave Pareto front.
- For the reward shaping baseline Calian, et al. consider quite large steps for the chosen weights. I expect when tuning those weights for the task the performance for this method will increase and therefore present a better baseline.
- The appendix also touches slight upon using multiple constraints. How would you expect your method to perform when introducing a larger number of constraints, which for some tasks can be easily possible.

---

> ### Author Response · Authors · 2021-08-24
> **Author Response (Part 2)**
>
> **Relabeling preferences:**
>
> We decided to relabel preferences based on prior work in off-policy learning that shows relabeling goals improves sample-efficiency of learning [2]. Relabeling preferences allows our algorithm to learn from off-policy data that is collected with a much different preference distribution than the current one. We have updated the paper to clarify this.
>
> We ran additional ablation studies with LP3 [MO-MPO-D] on the RWRL tasks, with and without preference relabeling (Appendix C.9). In this ablation study, preference relabeling improved performance in some but not all all tasks. Our hypothesis is that in a regime that requires more sample-efficient learning (e.g., learning from scratch on a real robot), relabeling preferences will have more benefit, because it makes it more feasible to learn from highly off-policy data.
>
> **Discussion of limitations:**
>
> We agree with the reviewer that an important limitation is LP3 [MO-MPO] and LP3 [MO-MPO-D] may not be sample-efficient enough to learn from scratch on physical robot systems. An alternative could be to first train a constraint-satisfying policy in simulation, and use this to accelerate learning on the real robot—e.g., train a policy on the real robot that maximizes reward while staying close to the simulation policy. This is an active and promising ongoing area of research (see [1] for a recent survey).
>
> There are several other limitations of this work. One main limitation is that in our experiments, we use a discrete distribution for the distribution over preferences. This means that the coverage of this distribution must include the ideal preference setting(s) that lead to constraint satisfaction. We made this implementation decision because it is interpretable and easy to train. In the future, it would be worthwhile to explore more flexible solutions.
>
> Another limitation is that LP3 focuses on policy improvement, and thus depends on accurate policy evaluation in order to perform well. As the reviewer mentioned, this is an issue in the Safety Gym point mass experiments, where the Q-function networks underestimate the Q-values for the obstacle collision cost. We hope to take advantage of future advances in policy evaluation to overcome this.
>
> Finally, because LP3 learns the preferences and policy in parallel, when learning a distribution over preferences in Module 2, it is important that the learning rate is not too high. If it is too high, then the preference distribution will quickly collapse to placing all probability on a small range of preferences, based on only initial learning of the policy—so this would not accurately capture which preferences actually lead to constraint satisfaction of the fully-trained policy.
>
> We have added a discussion of these limitations to the Conclusion.
>
> **Numbers reported in Calian et al.:**
>
> The numbers reported in Table 7 of Calian et al. are averaged across safety coefficients (of 0.05, 0.1, 0.2, and 0.3), but in our paper we consider a single safety coefficient per task (reported in Table 3). The safety coefficient is a parameter in the Real World RL suite that controls how difficult the constraint is to satisfy—the lower the safety coefficient, the harder it is to satisfy the constraints. For each task, we picked a safety coefficient from those four that was challenging but possible to satisfy.
>
> Table 7 of Calian et al. also does not report the cost per episode or the per-seed results. In order to compare against their approach and baselines, we obtained the raw results from the authors, and used those to compute the numbers in our Table I and generate the Pareto plots in Figure 8 in the Appendix.
>
> **Reward-shaping baseline:**
>
> We agree that the reward shaping baseline from Calian et al. could be better tuned. However, it is computationally expensive to search for the correct reward shaping weights for each constrained RL problem, because it requires training a separate policy per setting of weights. So in practice, this is not a feasible alternative for solving constrained RL problems. Since reward shaping is not a typical or feasible approach for constrained RL, we have removed that comparison from Table 1.
>
> **Typos:**
>
> We appreciate the reference and table corrections, and have fixed those in the updated paper.
>
> **Final thoughts:**
>
> Please let us know if this response and the updated paper address your concerns, and if not, what we can do to clarify any remaining questions.
>
> ---
>
> [1] Zhu et al. Transfer Learning in Deep Reinforcement Learning: A Survey. arXiv 2020.
>
> [2] Andrychowicz et al. Hindsight experience replay. NeurIPS 2017.

---

> > ### Comment · Reviewer_rQa3 · 2021-08-25
> > **Reviewer Response**
> >
> > I really appreciate it that you took the time to write this long response and clarified my open questions.
> > Overall, the explanations from the authors are sound, and I think the adjustments in the paper contribute to the understanding of the proposed method.
> > The discussion about limitations as well as the additional experiments and ablations further improve this work.
> > Hence, I believe that this paper is a good contribution to CoRL.
> >
> > I am looking forward to seeing more of this line of work in the future.

---

> ### Author Response · Authors · 2021-08-24
> **Author Response (Part 1)**
>
> We appreciate the detailed and thorough review. We are glad to see that the reviewer found the paper to be well-motivated, clear and well-organized, and has strong technical quality.
>
> **Choosing an appropriate trade-off:**
>
> We agree that choosing an appropriate trade-off is an important practical consideration. A natural approach is to use a coarse-to-fine grid search—one could condition the fully-trained policy on several preferences to investigate the resulting performance, then narrow down the search based on those results, and repeat. The learned preference distribution should only have non-zero probability on preferences that lead to constraint satisfaction (assuming policy evaluation during training is accurate), so deploying the policy should not lead to damaging outcomes.
>
> **Equality constraints with LP3 [MO-MPO-D]:**
>
> In fact, LP3 [MO-MPO-D] with equality constraints solves the same optimization problem as the Lagrangian relaxation version of the original optimization problem with inequality constraints. This is because Lagrangian relaxation introduces a duality gap, which turns the inequality constraint into an equality constraint.
>
> Another way of thinking about it is that this is the inductive bias that is introduced when learning a _single_ preference for Module 2. This effectively finds a solution that maximizes the allowed cost, so that it is equal to the constraint threshold
>
> **Toy Domain:**
>
> We re-ran all toy domain experiments, with five random seeds each for LP3 [LS] and LP3 [MO-MPO-D]. LP3 [LS] fails to find solutions on the concave Pareto front of Fonseca-Fleming, regardless of the seed choice. We updated Figure 2 with these results, and report the number of seeds and training details in Appendix B.1.
>
> **Scaling up to more constraints:**
>
> LP3 [MO-MPO-D] can easily scale up to problems with more constraints, in terms of computational cost. The additional computation cost comes from:
> * policy evaluation: a separate Q-function must be learned for each additional objective.
> * policy improvement: a separate improved action distribution $q_k$ must be learned for each additional objective.
>
> These can both be easily parallelized across objectives, to reduce the increase in wall-clock time. The impact on performance depends on the actual objectives themselves—intuitively, adding extra objectives that conflict with the existing objectives would make the problem harder and lead to a decrease in performance. Whereas adding extra objectives that align with the existing objectives may actually improve performance, by acting to regularize the policy, which is similar to the principle underlying auxiliary rewards.
>
> We agree that many real-world tasks may have multiple constraints. We ran an additional experiment with two constraints, on walker run from DeepMind Control Suite. The task reward is torso height, and we place inequality constraints on moving forward and energy usage (captured by action norm cost). In this task, LP3 [MO-MPO-D] is able to train preference-conditioned policies that capture a portion of the Pareto front. We added the results to the Experiments section.
>
> **Learning a set of policies seems seed-dependent:**
>
> When we use LP3 [CMO-MPO-D] to learn a set of constraint-satisfying policies for humanoid run, the results are indeed somewhat seed-dependent (Figure 4). This is because in humanoid run, the task reward is given for moving quickly _in any direction_, not necessarily forward. Thus there are a variety of running gaits that perform well across the objectives. The agent may learn any of these gaits, depending on the policy initialization. The ground-truth Pareto front likely consists of a mixture of these gaits. For instance, we observed that whirling is particularly effective at obtaining high task reward, whereas a human-like forward running gait does a good job at achieving moderate task reward with low action norm cost.
>
> In contrast, in the walker run task mentioned above, there is only one type of gait (moving forward) that performs well on the objectives, and as a result there is less variability in the performance of LP3 [CMO-MPO-D] across seeds (Figure 12).
>
> We also ran additional experiments with humanoid stand, and here there is only minor variability in the performance of LP3 [CMO-MPO-D] across seeds (Figure 11).
>
> In other words, this can be seen as a result of implementation, not because of the algorithm itself. Because LP3 [CMO-MPO-D] trains a _single_ policy network to capture a portion of the Pareto front, it is difficult to represent multiple strategies with this one policy. Instead, the policy initialization biases the network toward learning a particular strategy. So the learned portion of the Pareto front may differ depending on the seed, if there are a variety of strategies that can be used to solve a task. In future work, it would be interesting to instead train a collection of separate policy networks at once, for learning a wider variety of strategies.

---

### Official Review · Reviewer_Z5YJ · 2021-07-22

**Originality:** Very Good
**Technical Quality:** Very Good
**Clarity Of Presentation:** Very Good
**Impact:** 3

**Recommendation:**

Weak Accept: I recommend accepting the paper, but will not argue for my recommendation if the majority of other reviewers have a different opinion.

**Summary:**

This paper proposes a high-level framework called Learning Preferences and Policies in Parallel (LP3), which can derive new approaches basing on existing methods. Based on the idea of learning two modules separately, this paper solves the problem of constrained multi-objective reinforcement learning with better performance. The article has some innovative significance, and the experimental scheme is basically reasonable. What’s more, the experimental results and data in this paper are relatively good and reliable. In the open simulation environment, the RL system is verified. From the perspective of the algorithm, the verification is obtained, which makes a certain contribution to the development of Reinforcement Learning.

**Issues:**

1.Lines 105-106, for the traditional definition of non-dominated policy, the meaning of these two sentences in this position and the author's intention are not clear enough.

2.Lines 214-215, the reason for using uniform-discrete distribution is not clear enough.

3.The length of the basic knowledge in RL should be reduced, and the algorithm implementation and experimental results should be analyzed more. For those who are new to this professional field, this paper can also cite classic representative documents to convey basic theories.
4. From Figure 1 left, the framework composed of module 1 and module 2 gets a loop iteration, but in fact, only a limited number of combinations are made in the paper. Strictly speaking, this cannot be called a framework, in other words, the description of the framework in the paper is a little exaggerated.



**Reviewer Expertise:**

Very good: Comprehensive knowledge of the area

**Strengths And Weaknesses:**

This paper proposes the LP3 framework, considering the RL algorithm from the top-level perspective, which improves the efficiency of algorithm promotion, and at the same time has a positive significance for the development of RL. However，for the mathematical concepts and derivation of the traditional RL algorithm, the description in this paper is slightly cumbersome. What's more, the definition of the framework in the title of this paper is not reasonable, because the framework is not universally validated.

**Summary Of Recommendation:**

This article proposes a new RL insight (introducing learning preference), and tries to explain the idea with a general framework. However, the transferability of this framework to other Constrained Multi-Objective Reinforcement Learning problems remains to be verified. Therefore, it is recommended to reconsider the definition of the framework.

---

> ### Author Response · Authors · 2021-08-24
> **Author Response**
>
> We appreciate the review. We are glad to see that the reviewer found that this work is innovative, relevant for the community, and backed up by solid empirical evaluation.
>
> We have a couple questions for the reviewer, to better understand how we can address the concerns in the updated paper.
> * What aspects of the algorithm implementation and experimental results would benefit from more analysis?
> * What additional experiments would be useful for fully validating LP3 as a framework?
>
> **Limited number of instantiations of LP3 framework:**
>
> We call LP3 a framework because it is flexible in terms of which approach is chosen for Module 1 and Module 2. LP3 provides a basic structure for understanding existing constrained RL algorithms (i.e., Lagrangian relaxation), and for deriving new algorithms.
>
> We found it more intuitive to explain LP3 as a framework rather than as an approach or algorithm. This is because to obtain algorithms/approaches like LP3 [LS], LP3 [MO-MPO], or LP3 [MO-MPO-D], we have to instantiate LP3 with a particular choice for Module 1 and Module 2. That being said, if the reviewer is not satisfied with our reasoning, we are open to changing the naming from "framework" to a term that the reviewer thinks is more suitable.
>
> We agree that it would be worthwhile to instantiate LP3 with other options for Module 1. However, our focus is on continuous control and robotics domains, and unfortunately there are very few multi-objective RL algorithms that have been shown to work for continuous control and deep RL. We only know of MO-MPO [1], PG-MORL [2], and Chen et al. [3]. We could not use the latter two because they are not compatible with a preference or preference distribution that changes over the course of training. In future work, it is worth investigating how these two algorithms could be extended to allow for changing preferences.
>
> **Other clarity issues:**
>
> * When describing multi-objective MDPs (MO-MDPs), we define non-dominated policies (lines 105-106) because these are considered to be the optimal policies for a MO-MDP. Solving a MO-MDP thus means finding the set of all non-dominated policies, also known as the Pareto front. We have updated the paper to clarify this.
> * For learning a distribution of preferences, we use a discrete distribution with equally-sized bins. We made this implementation choice because it can easily represent probability distributions with multiple modes and arbitrary shapes, whereas a distribution like a Gaussian would not be able to. We initialize this distribution to uniform at the start of training, because we do not know a priori which preferences would lead to constraint-satisfying solutions. We have updated the paper to clarify this.
> * We agree that someone who is familiar with RL may find the background in this paper unnecessarily detailed. However, our aim is to make the paper as self-contained and accessible as possible, given the page limits. As suggested, we have added citations to a few classic papers for readers to delve deeper into the underlying theories.
>
> ---
>
> [1] Abdolmaleki et al. A distributional view on multi-objective policy optimization. ICML 2020.
>
> [2] Xu et al. Prediction-guided multi-objective reinforcement learning for continuous robot control. ICML 2020.
>
> [3] Chen et al. Meta-learning for multi-objective reinforcement learning. IROS 2019.

---

> > ### Comment · Reviewer_Z5YJ · 2021-09-02
> > **Reviewer Response**
> >
> > First of all, I’m willing to make some necessary comments to the two questions that the authors raised.
> >
> > 1.	Actually, in my last comment, although the issue in 1st one was not mentioned, it is indeed a nice topic for analysis, which may play an important role in keeping improving the theoretical level of this paper. So, as for ‘more analysis’, humanoid robot jumping control could be another task, and there are many ideas for more analysis.
> >
> > 2.	Reviewer thinks that the existing experiments in this paper have sufficient logical capabilities. However, reviewer is glad to communicate on the 2nd question raised by the authors. At least in my cognition, the word ‘framework’ may easily be understood as a universal method in a professional field. Of course, the effect of LP3 is good according to the current experimental results, but the effectiveness of this ‘framework’ does need to be verified for the new algorithms and new application scenarios discovered later (in compliance with LP3 requirements). Especially in the process of migrating theory to engineering, this is crucial.
> >
> > What the most important is the ‘framework’ is basically reasonable judging from the newly added results, while I recommend using other similar definition.
> >
> > Finally, it’s ok to change or not.

---

### Official Review · Reviewer_uAjH · 2021-07-23

**Originality:** Good
**Technical Quality:** Very Good
**Clarity Of Presentation:** Very Good
**Impact:** 3

**Recommendation:**

Weak Accept: I recommend accepting the paper, but will not argue for my recommendation if the majority of other reviewers have a different opinion.

**Summary:**

This work explores two problems in RL, learning a policy with constraints and multi-objective learning. Currently, known methods exist to solve each of these problems.  The authors merge solutions to these two problems so that constraints can be placed on a multi-objective MDP.  To do this, the authors define an MDP with multiple objectives, and each objective can have a constraint on it, which they title a "constrained multi-objective MDP".  The method is then derived which combines known work on multi-objective policy optimization (MO-MPO) and learning with constraints to solve this class of MDPs.  The main contribution of this work is that the approach is able to approximate the Pareto front of optimal policies by learning a segment of the front that satisfies the given constraints.  This presents the benefit of finding Pareto optimal policies that balance between satisfying constraints and task reward.

**Issues:**

Mainly concerning the novelty over MO-MPO.

**Reviewer Expertise:**

Good: General knowledge of the area

**Strengths And Weaknesses:**

The approach of this work is quite intuitive, and the writing is clear and convincing. This work derives intuitive connections between the constrained optimization and the Pareto front.  The problem of constrained multi-objective learning is highly relevant to robot learning and could make a good contribution to CoRL.

A key strength of this paper is that the results achieved appear to be significant over baselines and the experimental evaluation is quite extensive.  I feel that this work will provide a useful contribution to the set of RL tools and will likely be used.  The clarity of this paper is also good considering the wide range and complexity of methods discussed.  The writing is well polished.

My main issue is that the method in this paper feels incremental compared to MO-MPO. The authors apply well known methods from constrained policy optimization to the multi objective framework.   The connections found between constrained optimization and the Pareto front are well known in other multi-objective problems but have not been explored in RL, to my knowledge.  Overall, this does help improve the novelty.

**Summary Of Recommendation:**

I am learning towards recommending acceptance of this work and think that it will be useful to the audience of CoRL.  The problem is highly relevant, and the connections made are intuitive.  The evaluation of the method is convincing and the writing is clear.

---

> ### Author Response · Authors · 2021-08-24
> **Author Response**
>
> We appreciate the review. We are glad to see that the reviewer thought the writing is clear and convincing, the approach is intuitive, the problem setting is highly relevant to robot learning, and the empirical evaluation is extensive and shows clear improvement over baselines.
>
> **Novelty with respect to MO-MPO:**
>
> Our paper introduces a novel application of multi-objective RL to constrained RL problems, that leads to better results.
>
> While MO-MPO focuses on multi-objective RL, this paper focuses on constrained RL problems. MO-MPO cannot be directly applied to solve such problems, because the preference is fixed throughout training. We do not know a priori which preference settings would lead to policies that satisfy the constraints, so we would need to resort to figuring this out via trial and error.
>
> We consider LP3 [MO-MPO] and LP3 [MO-MPO-D] as novel instantiations of LP3 because this is the first time MO-MPO has been successfully applied to constrained RL problems. This is possible because LP3 proposes a way to _learn_ the preferences over training, that leads to learning constraint-satisfying policies.  In other words, to the best of our knowledge, there are no existing algorithms _for constrained RL_ that are similar to LP3 [MO-MPO] or LP3 [MO-MPO-D].
>
> In addition, we go beyond well-known approaches in constrained policy optimization. With LP3 [MO-MPO-D], we explore learning a _set_ of constraint-satisfying policies that make different tradeoffs between the reward and cost. This is useful when the exact desired constraint threshold is not known a priori, or if the constraint is a soft constraint; this is common in real-world problems including robotics. For instance, perhaps we want a robot to do a task efficiently while minimizing impact forces, but we do not know exactly how much impact force would cause wear-and-tear to the robot.
>
> Approaches for solving constrained MDPs cannot handle unknown constraint thresholds or soft constraints, because they search for a single constraint-satisfying solution that maximizes the reward. This is why we formalized the problem as a constrained multi-objective MDP, which is what LP3 [MO-MPO-D] solves.
>
> **Final thoughts:**
>
> Please let us know if this response addresses your concerns, and if not, what we can do to clarify any remaining questions.

---

### Official Review · Reviewer_YksG · 2021-07-24

**Originality:** Good
**Technical Quality:** Good
**Clarity Of Presentation:** Very Good
**Impact:** 4

**Recommendation:**

Weak Accept: I recommend accepting the paper, but will not argue for my recommendation if the majority of other reviewers have a different opinion.

**Summary:**

The authors present a more generalized framework for handling multi-objective RL problems which merges concepts of both constrained MDPs and multi-objective MDPs with in a circular two-phase optimization where the RL policy is optimized for a set of preferences followed by the preferences being optimized for a set of constraints and repeating. The authors also claim that the resulting LP3 is more sample efficient, able to find sets of viable policies and able to handle soft constraints.

**Issues:**

Experiments with more than 2 objectives would be the major thing I'd hope to see before I can accept this as a 'multi' objective solution

**Reviewer Expertise:**

Very good: Comprehensive knowledge of the area

**Strengths And Weaknesses:**

### Strengths:

S1) The paper has generally decent flow and communicates most of its ideas clearly with sufficient background

S2) I always appreciate the process of starting demonstrations with toy problems and working up to more complex domains as it does better to establish the viability of an approach in more intuitive terms.

S3) I like the idea of addressing soft constraints or constraint bounds as opposed to strict constraints as this can relate to more realistic scenarios.

S4) The website has intuitive visualizations that help drive the key points of the paper home.

### Weaknesses:

W1) It appears to me that, despite talking about a 'multi-objective' setting, many of the experiments relate mainly to just a single reward and a single policy regularization constraint - and even that seems mostly related to actuation costs. How would this method extend to other types of objectives, and how well would the algorithm handle a more than two soft constraints - what are the associated performance and computational tradeoffs? I feel like these are important questions to address that are not addressed in the paper.

W2) I am not convinced that treating MO-MPO as a special module really constitutes a novel instantiation of LP3 in a significantly meaningful sense (at least not in a way that I'd regard a novel module) - unless I am mistaken, the extension proposed in this paper is mostly just input-state augmentation, am I wrong?

W3) It would help if the figure captions were more self-contained and descriptive, with important takeaways summarized. The inconsistency of the formatting for the caption for Figure 2 also interrupted the flow of reading.

### General notes:

Overall, I think the methods proposed here have promise, but need more thorough evaluation to demonstrate broader value. One possible exploration could be in looking at how the different reward components for the various tasks could be satisfied (for example, the gym mujoco control tasks have rewards associated with joint limits, energy consumption, task progress, etc. - so considering them all separately as constraints could make for interesting test cases)

**Summary Of Recommendation:**

While I think the work shows promise, I am as yet unconvinced of broader utility by the experimental results.

(Updated after rebuttal):
I am more convinced, given the additional results presented and additional contextualization provided. I am still not convinced on the issue of the novelty of MO-MPO, but I am willing to let it slide given that no other major concerns are raised

---

> ### Author Response · Authors · 2021-08-24
> **Author Response (Part 1)**
>
> We appreciate the detailed review. We are glad to see that the reviewer found the paper was clear, the systematic evaluation of the approach was valuable, and being able to handle soft constraints is relevant for real-world problems.
>
> **Experiments with more than two objectives:**
>
> Although our evaluation is primarily on problems with two objectives, prior work on multi-objective RL focuses on this setting as well. This includes recent approaches on deep RL for multi-objective continuous control tasks: MO-MPO [1], PG-MORL [2], and Chen et al. [3].
>
> We agree, however, that it is useful to validate LP3 on problems with more than two objectives. In our original Appendix C.6, we have an experiment with three objectives, on the Safety Gym point goal task. The three objectives are to reach the goal, avoid crossing hazards on the floor, and avoid bumping into vases. We trained policies for a selection of constraint thresholds on the last two objectives. The results are similar to the results from the two-objective experiments: we find that LP3 [MO-MPO-D] outperforms the LP3 [LS] baseline when the constraints are difficult to satisfy. In this point goal experiment, scaling up from two to three objectives did not noticeably affect the sample efficiency of LP3 [MO-MPO-D].
>
> We also ran a new experiment with three objectives, on walker run from DeepMind Control Suite. The objectives are torso height, moving forward, and energy consumption (captured by action norm cost, as is done in the OpenAI Gym Mujoco tasks). We place inequality constraints on moving forward and energy consumption. In this task, LP3 [MO-MPO-D] is able to train preference-conditioned policies that capture a portion of the Pareto front. We have added these two experiments with three objectives at the end of the Experiments section of the main paper.
>
> LP3 [MO-MPO-D] can easily scale up to problems with more constraints, in terms of computational cost. The additional computation cost comes from:
> * policy evaluation: a separate Q-function must be learned for each additional objective.
> * policy improvement: a separate improved action distribution $q_k$ must be learned for each additional objective. Note that this does not require training a separate neural network to estimate $q_k$; it only requires an extra temperature parameter (see Appendix E.1).
>
> These can both be easily parallelized across objectives, to reduce the increase in wall-clock time. The impact on performance depends on the actual objectives themselves—intuitively, adding extra objectives that conflict with the existing objectives would make the problem harder and lead to a decrease in performance, because it is harder to find a policy that does well on all objectives. Whereas adding extra objectives that align with the existing objectives may actually improve performance, by acting to regularize the policy, which is similar to the principle underlying auxiliary rewards. We have added a discussion of these points to the Conclusion.
>
> **Considering other types of objectives:**
>
> In our empirical evaluation, we purposely chose constrained RL problems that cover a variety of objectives, that reflect the kinds of objectives that one might find in robotics problems.
>
> Our experimental domains cover the following costs:
> * Humanoid: energy usage, which is captured by actuation cost
> * Real World RL: balance velocity, joint angle, or joint velocity, depending on the task
> * Safety Gym and Robot: collision avoidance with obstacles (both static and dynamic obstacles are considered)
>
> These domains also cover different types of task reward:
> * Humanoid and Real World RL: dense locomotion-based task reward
> * Safety Gym: sparse task reward for reaching a goal or pushing a block to a goal, depending on the task
> * Robot: sparse task reward for pushing a block to a goal
>
> ---
>
> [1] Abdolmaleki et al. A distributional view on multi-objective policy optimization. ICML 2020.
>
> [2] Xu et al. Prediction-guided multi-objective reinforcement learning for continuous robot control. ICML 2020.
>
> [3] Chen et al. Meta-learning for multi-objective reinforcement learning. IROS 2019.

---

> ### Author Response · Authors · 2021-08-24
> **Author Response (Part 2)**
>
> **Novelty of LP3 versus MO-MPO:**
>
> We consider LP3 [MO-MPO] and LP3 [MO-MPO-D] as novel instantiations of LP3 because this is the first time MO-MPO has been successfully applied to constrained RL problems. This is possible because LP3 proposes a way to _learn_ the preferences during training, that leads to learning constraint-satisfying policies. In contrast, both MO-MPO and preference-conditioned MO-MPO cannot be directly applied to constrained RL problems, because the preference or preference distribution is fixed throughout training.
>
> Our empirical evaluation shows that LP3 [MO-MPO] and LP3 [MO-MPO-D] outperform existing constrained RL algorithms by a margin. We believe there is also novelty in showing that something existing works very well in a different context (in our case, constrained RL), and outperforms existing methods in that context.
>
> It is correct that the extension to MO-MPO that we propose (i.e., preference-conditioned MO-MPO), is mainly input-state augmentation of the Q-function and policy networks. The other algorithmic difference is that the closed-form solution for the improved action distributions $q_k$ must also now depend on the preference (see lines 751-756 in Appendix E.1).
>
> In other words, although Module 1 is not entirely novel by itself, we believe the overall algorithms of LP3 [MO-MPO] and LP3 [MO-MPO-D] are new—to the best of our knowledge, there are no existing algorithms _for constrained RL_ that are similar to LP3 [MO-MPO] or LP3 [MO-MPO-D].
>
> **Clarity of figure captions:**
>
> We appreciate the suggestions, and have updated the figure captions accordingly, with the aim of giving a reader a clear overview of the paper just by reading the captions. We moved the caption of Figure 2 to be left-aligned, to hopefully improve the flow of reading. We had to keep it as a side caption in order to stay within the page limit.
>
> **Final thoughts:**
>
> Please let us know if this response and the updated paper address your concerns, and if not, what we can do to clarify any remaining questions.

---

> > ### Comment · Reviewer_YksG · 2021-08-31
> > **Response to authors**
> >
> > Thank you for addressing the concerns raised. I shall use this comment to address both parts 1 and 2.
> >
> > The additional data and improvements to the writing clarity go a long way towards improving the presentation of this paper. I especially appreciate the efforts taken to demonstrate the utility of the proposed method on additional tasks with more objectives - while I concede to the authors that there is in fact precedent for just 2 objectives being treated as a full 'multi' objective system, I would have raised similar concerns had I been reviewing those papers too, so I appreciate that the authors of this paper added additional experiments.
> >
> > Strictly speaking, I am still not convinced on the point of treating MO-MPO as a 'novel' module (though clearly the distinction between instantiations makes a practical difference). However, if none of the other reviewers raise further concerns on the issue, I'm willing to let the point slide.
> >
> > Overall, I am significantly happier with the current state of the paper and can see it as a useful addition to research in this area.

---

### Author Response · Authors · 2021-08-27
**Author update: Exciting new results on a task with 4 objectives**

We are excited to present new results in a domain with _four_ objectives, based on the Control Suite humanoid walk task described in the paper. In additional to the task reward and energy usage objectives, we add objectives for moving forward and moving left. (Note that the original task reward is given for moving in _any_ direction.) This task is substantially more difficult than the planar walker task with three objectives that we added earlier this week: the action space is 21-dimensional for humanoid, compared to 6-dimensional for walker, and the humanoid is not constrained to planar movement.

We use LP3 [MO-MPO-D] to train policies for this more challenging task, with constraints on energy usage and either the move-forward or move-left objective. The fully-trained policies capture a portion of the Pareto front, with consistent performance across random seeds (Figure 13 in the Appendix). Sample efficiency is similar, despite having four total objectives and two constraints: these policies were trained for 200 million actor steps. In the paper, for the two-objective experiments with humanoid run and walk, we trained policies for 200-500 million actor steps.

What is particularly exciting is that the fully-trained policies learn to smoothly interpolate between walking forward and walking left, by varying the input preference setting. We added videos to the website (at [this link](https://sites.google.com/corp/view/cmorl/set-of-policies#h.8kbr5mwnf59g)) that show this behavior. In this task, walking sideways is a challenging gait to learn, because of difficulty balancing, and it is not a gait that would be likely to emerge naturally without placing a constraint on the move-left objective to be greater than a threshold. In addition, the constraint on action norm leads to more human-like gaits.

We hope that these experiments, in a complex continuous control domain with four objectives, help further validate the LP3 framework and the LP3 [MO-MPO-D] algorithm. These newer edits are in red in the main paper and Appendix.

---

### Meta-Review · Area_Chair_YKsj · 2021-08-14

**Recommendation:** Accept (Poster)
**Confidence:** 4

**Metareview:**

This paper proposes one general framework which can deal with the constrained multi-objective RL problems. Reviewers own the consistent comments that this framework is promising and useful in some sense. Experimental results are also convincing. However, there are still some concerns raised by reviewers, which need to be well answered to make this work much more qualified. For example, multi-objective setting in the experiments sounds like not so complex and general. Some empirical tuning and algorithmic design should be presented more clearly.

Although there are still some minor issues in this paper, most of reviewers think authors have addressed the majority of the concerns during the rebuttal period and it has met the quality of CoRL paper.

---

> ### Author Response · Authors · 2021-08-24
> **Author Response**
>
> We are glad that the reviewers believe our proposed framework makes a valuable contribution and is supported by convincing experimental results. We appreciate the reviewers' suggestions on how to strengthen the paper. Based on these suggestions, we have updated the paper to clarify the design of the algorithm and to include several new experiments. These changes are in blue in the updated version, to make them easily visible.
>
> In particular, we added another experiment with three objectives, that shows LP3 [MO-MPO-D] can learn a portion of the Pareto front with respect to _two_ objectives with constraints (Figure 5, right). We also added an ablation study on preference relabeling, re-ran experiments on the toy domain, and ran an extra experiment with LP3 [MO-MPO-D] on humanoid walk to investigate consistency across seeds. We hope that these additional experiments address the concerns raised by the reviewers.

---

### Decision · Program_Chairs · 2021-09-13

**Decision:**

Accept (Poster)

**Comment:**

This paper proposes one general framework which can deal with the constrained multi-objective RL problems. Reviewers own the consistent comments that this framework is promising and useful in some sense. Experimental results are also convincing. However, there are still some concerns raised by reviewers, which need to be well answered to make this work much more qualified. For example, multi-objective setting in the experiments sounds like not so complex and general. Some empirical tuning and algorithmic design should be presented more clearly.

Although there are still some minor issues in this paper, most of reviewers think authors have addressed the majority of the concerns during the rebuttal period and it has met the quality of CoRL paper.